# In Vitro and In Vivo Effect of Amikacin and Imipenem Combinations against Multidrug-Resistant *E. coli*

**DOI:** 10.3390/tropicalmed7100281

**Published:** 2022-10-02

**Authors:** Sara Mahmoud Farhan, Rehab Mahmoud Abd El-Baky, Salah Abdalla, Ahmed Osama EL-Gendy, Hala Rady Ahmed, Doaa Safwat Mohamed, Amr El Zawily, Ahmed Farag Azmy

**Affiliations:** 1Department of Microbiology and Immunology, Faculty of Pharmacy, Beni-Suef University, Beni-Suef 62514, Egypt; 2Department of Microbiology and Immunology, Faculty of Pharmacy, Deraya University, Minia 11566, Egypt; 3Department of Microbiology and Immunology, Faculty of Pharmacy, Minia University, Minia 61519, Egypt; 4Department of Microbiology and Immunology, Faculty of Pharmacy, Suez Canal University, Ismailia 41522, Egypt; 5Department of Microbiology and Immunology, Faculty of Pharmacy, Sohag University, Sohag 82524, Egypt; 6Department of Plant and Microbiology, Faculty of Science, Damanhour University, Damanhour 22511, Egypt; 7Department of Pharmaceutical Sciences and Experimental Therapeutics, College of Pharmacy, University of Iowa, Iowa City, IA 52242, USA

**Keywords:** *E. coli*, drug combination, imipenem, amikacin, synergism, Metallo-β-lactamase gene Imipenemase (*bla-_IMP_*), aminoglycoside 6′-N-acetyltransferase (*aac (6′)-Ib*) genes, SEM

## Abstract

Introduction: The emergence of multidrug-resistant (MDR) *E. coli* has developed worldwide; therefore, the use of antibiotic combinations may be an effective strategy to target resistant bacteria and fight life-threatening infections. The current study was performed to evaluate the in vitro and in vivo efficacy of amikacin and imipenem alone and in combination against multidrug-resistant *E. coli.* Methods: The combination treatment was assessed in vitro using a checkerboard technique and time-killing curve and in vivo using a peritonitis mouse model. In resistant isolates, conventional PCR and quantitative real-time PCR techniques were used to detect the resistant genes of Metallo-β-lactamase gene Imipenemase (*bla-_IMP_*) and aminoglycoside 6′-N-acetyltransferase (aac *(6′)-Ib*). Scanning electron microscopy was used to detect the morphological changes in the resistant isolates after treatment with each drug alone and in combination. In vitro and in vivo studies showed a synergistic effect using the tested antibiotic combinations, showing fractional inhibitory concentration indices (FICIs) of ≤0.5. Regarding the in vivo study, combination therapy indicated a bactericidal effect after 24 h. *E. coli* isolates harboring the resistant genes Metallo-β-lactamase gene Imipenemase (*bla-_IMP_*) and aminoglycoside 6′-N-acetyltransferase (*aac (6′)-Ib*) represented 80% and 66.7%, respectively, which were mainly isolated from wound infections. The lowest effect on Metallo-β-lactamase gene Imipenemase (*bla-_IMP_*) and aminoglycoside 6′-N-acetyltransferase (*aac (6′)-Ib*) gene expression was shown in the presence of 0.25 × MIC of imipenem and 0.5 × MIC of amikacin. The scanning electron microscopy showed cell shrinkage and disruption in the outer membrane of *E. coli* in the presence of the antibiotic combination. Amikacin and imipenem combination can be expected to be effective in the treatment and control of serious infections caused by multidrug-resistant (MDR) *E. coli* and the reduction in bacterial resistance emergence.

## 1. Introduction

*Escherichia coli* is a rod-shaped Gram-negative facultative anaerobe bacterium found in the microbiota of human and animal digestive tracts [1]. *E. coli* pathotypes are involved in hospital-acquired pneumonia (HAP), urinary tract infections (UTI), gastrointestinal tract infections, surgical site infections (SSI), sepsis, meningitis, and hemolytic–uremic syndrome (HUS) [2]. The genetic elements of *E. coli* can be transferred horizontally between related bacteria or bacteria from different families [2]. *E. coli* is one of the important species causing infections that are characterized by high morbidity and mortality rates worldwide (about 8.5 deaths per 100,000 persons–year) [3,4]. Multidrug-resistant *E. coli* is defined as those strains that are resistant to one or more antibiotics in three or more antimicrobial categories [5]. The wide spread of resistance limits the therapeutic activity of last resort, efficacious antibiotics [6]. 

Carbapenems are considered as last resort antibiotics that can treat drug-resistant Gram-negative bacteria. The molecular structure of carbapenem together with the beta-lactam ring show high stability against most beta-lactamases [7]. Due to uncontrollable public access to antibiotics and the use of subtherapeutic doses, carbapenem resistance is reported. In addition, aminoglycoside antibiotics confer bactericidal activity in emerging systemic infections [8], as they target the bacterial ribosome causing the inhibition of protein biosynthesis [9]. Aminoglycoside-modifying enzymes and RNA methyltransferases are the main mechanisms of aminoglycoside resistance commonly encountered in carbapenem-resistant isolates [10].

Carbapenemase enzyme production is the main cause of carbapenem resistance [11], and they are classified into three groups: Ambler [12] Class A (clavulanic acid-inhibitory extended-spectrum β-lactamases) *Klebsiella pneumoniae* carbapenemase (KPC); Class B Metallo-β-lactamases (MBLs) that include New Delhi MBL (NDM), Verona integrin-encoded MBL (VIM), and imipenemase (IMP); and Class D oxacillinases (OXA)-type enzymes which include OXA-48-like carbapenemases. These enzymes showed variable levels of resistance among carbapenem antibiotics [13]. Limited choices are available in the treatment of carbapenem-resistant Gram-negative bacteria; therefore, there is an urgent need for new effective antimicrobials [7]. Many studies have reported the effectiveness of antibiotic combinations against multidrug-resistant (MDR) strains in comparison to the use of a single antibiotic [14,15]. 

Aminoglycosides are reported to be a preferred agent for combining antibiotics because of their wide spectrum and their good synergistic activity with antibiotics that affect the bacterial cell wall, such as β-lactams, glycopeptides, monobactam, and carbapenems [16,17,18,19]. Previous studies revealed that the synergistic activity among β-lactams and amikacin was not attributed to the level of the minimum inhibitory concentrations (MICs) of the antibiotics tested in combination but due to the post-antibiotic effect (PAE) of the tested amikacin and other antibiotics [20,21]. The synergistic effect of amikacin with other antimicrobials was studied by many researchers that explained the action of combinations tested based on post-antibiotic effects [20,21,22,23]. 

In our hospitals, imipenem and amikacin are widely prescribed antibiotics. Several studies reported the wide spread of aminoglycoside resistance mobilized with β-lactamase genes carrying integrons except for *bla-_SPM_*. Depending on the previous findings, our study aimed to evaluate the in vitro and in vivo activities of amikacin and imipenem as a monotherapy and in combination against multidrug-resistant *E. coli*.

## 2. Materials and Methods

### 2.1. Culture and Bacterial Identification

A total of two hundred clinical samples were collected from patients admitted to intensive care units of Minia University Hospital (Minia, Egypt) suffering from chest, wound, burn, urinary tract, and ear infections, and gastroenteritis. Their age ranged from 23 to 71 years: 67% (134/200) were male, and 33% (66/200) were female. One hundred and fifty nonrepetitive isolates were Gram-negative bacteria identified using traditional methods. *E. coli* was the most prevalent bacteria (60 isolates) followed by *Pseudomonas. aeruginosa* (45 isolates), *Proteus* spp. *(*30 isolates), *Klebsiella* spp. (10 isolates), and *Acinetobacter baumannii* (5 isolates). The need for patient consent was waived by the ethics committee. Samples were processed and grown on trypticase soy agar (Lab M, Heywood, UK) at 37 °C overnight. *E. coli* colonies showed a pink color on MacConkey agar and on eosin methylene blue (EMB) (Lab M, Heywood, UK), appeared as a green metallic sheen. Colonies were further identified using traditional microbiological and biochemical tests [24].

### 2.2. Antibiotic Susceptibility Tests

Antibiotic sensitivity tests were performed using the Kirby–Bauer disk diffusion method with cefpodoxime (30 µg), tetracycline (30 µg), amikacin (30 µg), cefadroxil (30 µg), streptomycin (10 µg), aztreonam (30 µg), ceftriaxone (30 µg), cephalothin (30 µg), gentamycin (10 µg), amoxicillin/clavulanic acid (30 µg), ceftazidime (30 µg), imipenem (10 µg), meropenem (10 µg), cefoperazone (75 µg), doxycycline (30 µg), ciprofloxacin (5 µg), nalidixic acid (30 µg), cefotaxime (30 µg), piperacillin (100 µg), cefepime(30 µg), ampicillin/sulbactam (20 µg), norfloxacin (10 µg), tobramycin (10 µg), sulphamethoxazole/trimethoprim (25 µg), nitrofurantoin (300 µg), chloramphenicol (30 µg), levofloxacin (5 µg), piperacillin/tazobactam (10 µg), ofloxacin (10 µg), and azithromycin (30 µg). Zones of inhibition were determined according to CLSI 2018 [25]. Antibiotic susceptibility was further confirmed for resistant *E. coli* using minimum inhibitory concentrations (MICs) for both imipenem (IMP) (MIC, R ≥ 4 µg/mL) and amikacin (AK) (MIC, R ≥ 64 µg/mL) confirmed by broth microdilution according to the Clinical and Laboratory Standards Institute 2018 [25].

### 2.3. PCR Detection of Metallo-β-Lactamase Gene Imipenemase (bla-_IMP_) and Aminoglycoside 6′-N-acetyltransferase (aac (6′)-Ib) in Selected Resistant Bacteria

The DNA template was extracted by boiling (10 min at 95 °C) [26], and PCR was performed in a thermal cycler (Bio Rad, USA) using 25-μL volumes including 12.5 μL of PCR Master Mix (500 mM of Tris-HCl pH 8.55, 1.5 mM of MgCl2, 0.2 mM of dNTPs, and 0.04 units/uL of Taq DNA polymerase). PCR primers and PCR conditions are presented in Table 1. After amplification, 10 μL of each sample was analyzed with 2% agarose gel electrophoresis for the detection of positive samples. A 1000-bp DNA ladder was used to detect the product sizes of the genes Metallo-β-lactamase gene Imipenemase (*bla_IMP_*) (488 bp) and aminoglycoside 6′-N-acetyltransferase (*aac (6′)-Ib*) (365 bp).

### 2.4. Checkerboard Assay

The combined effect of both amikacin and imipenem were tested against clinical resistant *E. coli* isolates to determine the fractional inhibitory concentration (FIC) indices. FIC index is defined as the ratio between MIC alone and MIC in combination as follows: 

FIC index = FIC of drug A + FIC of drug B.

FIC of drug A = MIC of drug A in combination/MIC of drug A alone.

FIC of drug B = MIC of drug B in combination/MIC of drug B alone.

Results were interpreted as follows: synergy showed as FIC index ≤ 0.5; antagonism showed as FIC index of ≥2; and additive showed as FIC index of >0.5 but ≤1 [17,30].

### 2.5. Time-Kill Analysis

Time-kill curves were performed at concentrations of 1/2× MIC, 1× MIC, 2× MIC, and 4× MIC for each tested antibiotic alone or in combination. Three representative isolates were chosen: one isolate was resistant to both antibiotics; one isolate was resistant to imipenem only; and the other isolate was resistant to amikacin only. They were chosen based on the checkerboard results. Bacterial counts were detected after 0, 2, 4, 8, and 24 h after incubation at 37 °C. The limit of detection was 300 CFU/mL. Synergy was defined as a 2 log_10_ decrease in the colony counts caused by the combination compared to the most active single drug at single time-points; additivity was a >1 log_10_ but <2 log_10_ decrease. Antagonism was defined as a >2 log_10_ increase in colony count caused by the combination compared to that by the most active antibiotic alone at any time-point [17]. Bacteriostatic activities were defined as the presence of ≥2 log_10_ but <3 log_10_ reductions and bactericidal activities as the presence of ≥3 log_10_ reductions in CFU/mL at 24 h compared to the initial inoculum [31].

### 2.6. In Vivo Study

Specific pathogen-free ICR (female) mice at 6 weeks old, weighing 30*–*38 g (National Research Center, Dokki, Giza, Egypt) were used in this study. The mice were divided into 4 groups, with 8 mice in each group. The mice were intraperitoneally infected with (0.5 MacFarland 1.5 *×* 10^8^ CFU/mL; 0.2 mL) *E. coli* resistant to both drugs.

Animals were kept in well-ventilated cages at temperature of 25 ± 2 °C, humidity of 60 ± 10%, and normal photoperiod 12/12 h light–dark cycles under stress-free conditions with free access to food and water. Animals were observed every day for changes in body weight or any other clinical signs of bacterial infections, such as change in body posture and coat appearance, a decline in activity, and changes in food and drink intake. No specific housing requirements were provided. All the staff that handled, administered medications to, and took care of the animals were properly trained. 

Animals were treated for 24 h, starting 3 h after bacterial challenge. Imipenem was administered intraperitoneally with 40 mg/kg every 4 h. Amikacin was administered 15 mg/kg every 8 h, and amikacin and imipenem in combination (doses and intervals were the same as in monotherapy). The whole experiment took 27 h. Four groups of animals were evaluated and inoculated with the *E. coli* resistant to both drugs. Group I: Normal control group were divided into two subgroups: the negative control group received intraperitoneal sterile saline, and the positive and tested group received intraperitoneal bacteria. (Humane endpoint was determined through overdose of thiopental.) Test groups received intraperitoneal bacteria and were treated as follows: Group II: tested group received amikacin dose (200 µL intraperitoneal) with 15 mg/kg every 8 h; Group III: tested group received imipenem dose (200 µL intraperitoneal) with 40 mg/kg every 4 h; and Group IV: tested group received intraperitoneal combination of amikacin and imipenem the same as a single dose. 

Twenty µL blood samples were taken from the tail vein of 5 mice randomly selected in each group at 3, 11, and 27 h after infection. Seven two-fold serial dilutions were plated on Mueller–Hinton agar plates by adding 10 microliter aliquots and incubated overnight at 37 °C for CFU determination [32]. The study protocol conformed to the ethical guidelines of the 1975 Declaration of Helsinki, as revealed in a priori approval (8/2021) by ethical review board of Faculty of Pharmacy, Deraya University, Egypt.

### 2.7. Gene Expression of Metallo-β-Lactamase Gene Imipenemase (bla-_IMP_) and Aminoglycoside 6′-N-acetyltransferase (aac (6′)-Ib) Using Real-Time PCR

RNA was isolated from log phase cells of *E. coli* resistant strains to both imipenem and amikacin exposed to sub-MIC. Gene expression for *bla-_IMP_* and *aac(6′)-Ib* with 16S rRNA house-keeping gene as a control gene in the resistant isolate (isolate W3) was performed before and after treatment with the antibiotic.

Bacterial RNA was extracted using QuantiTect SYBR Green PCR kit (Qiagen, Hilden, Germany). The RT-PCR was performed in 25 μL reaction mixture consisting of 2× QuantiTect SYBR Green PCR Master Mix (12.5 μL), reverse transcriptase (0.25 μL), 0.5μL of each forward (20 pmol) and reverse (20 pmol) primers, RNase-free water (8.25 μL), and template RNA (3 μL). The cycling conditions are indicated in Table 1. Strata gene MX3005P software was used for the detection of amplification curves and CT values. The variation of gene expression on the RNA of the different samples was estimated. CT of each sample was compared with that of the control group according to the “ΔΔCT” method stated by Yuan, et al. [33]. Dissociation curves were compared between different samples for exclusion of false positive results.

### 2.8. Scanning Electron Microscopy (SEM)

*E. coli*-resistant colonies at the desired growth phase were suspended in a saline solution containing 0.2% Tween 80 and incubated at 37 °C with imipenem alone, amikacin alone, and in combination. The bacterial cells were centrifuged after 24 h at 8000 rpm for 15 min. The bacterial cells were then washed with 0.1 mol/L tris-acetate buffer (pH 7.1), fixed in tris-acetate buffer containing 1.5% glutaraldehyde, and then freeze-dried. Each of the bacterial cultures were observed by SEM (Hitachi, Japan) at magnifications of 7500, 10,000, and 20,000×. The bacterial suspension in saline with no antibiotics acted as a negative control [34].

### 2.9. Statistical Analysis

SPSS program (Statistical Package for Social Sciences) software version 25 was used to evaluate the data that had been statistically obtained. The graphic presentation was performed with the help of Microsoft Office Excel 365 software. For parametric (normally distributed) quantitative data, descriptive statistics were performed using the mean and standard deviation (SD). Shapiro–Wilk test was used to distribute the data. Analyses were carried out for parametric quantitative data between various groups or concentrations using the one-way ANOVA test, followed by the post hoc Tukey’s analysis between various groups or concentrations. The repeated measures ANOVA test was used to analyze parametric quantitative data between different periods, and it was followed by a post hoc LSD analysis between each pair of times. The significance level was set at (*p* value 0.05).

## 3. Results

### 3.1. E. coli Isolation and Identification

A total of 150 clinical nonrepetitive isolates of Gram-negative bacteria were identified using traditional methods. *Escherichia coli* was the most common Gram-negative bacteria, representing 40% of the total isolates (60/150).

### 3.2. Antibiotic Susceptibility Testing

High resistance to most of the tested antibiotics was shown, including amoxicillin/clavulanic acid, cefpodoxime, cefepime, imipenem, and others. On the other hand, piperacillin/tazobactam and levofloxacin were the most effective antibiotics.

*E. coli* isolates were found to be multidrug-resistant (MDR) where they were resistant to one or more agents in three or more antibiotic categories (Figure 1) (Appendix A). 

### 3.3. Detection of MIC_50_ and MIC_90_ of Imipenem and Amikacin for E. coli Isolates

The MIC values indicated that 75% and 58.3% of isolates were resistant to amikacin (MIC ≥ 64 µg/mL) and imipenem (MIC ≥ 4 µg/mL), respectively. Imipenem MIC_90_ and MIC_50_ were 512 µg/mL and 16 µg/mL, while amikacin MIC_90_ and MIC_50_ were 512 µg/mL and 64 µg/mL. 

### 3.4. Molecular Identification of Metallo-β-Lactamase Gene Imipenemase (bla-_IMP_) and Aminoglycoside 6′-N-acetyltransferase (aac (6′)-Ib)-Resistant Genes Using Conventional PCR

*E. coli* harboring bla-_IMP_ had 48 isolates (80%) of the total number of *E. coli*. bla-_IMP_ was commonly found among strains isolated from wound infections (47.9%), followed by patients admitted to intensive care units and from chest infections (12.5%), urinary tract infections (10.4%), gastroenteritis (8.3%), burn infections (6.3%), and ear infections (4.2%). Forty isolates (66.7%) of *E. coli* were positive for *aac(6′)-Ib*. *aac(6′)-Ib* was found to be the most common among wound infections (47.5%), followed by chest infections (15%), patients admitted to intensive care units (12.5%), burn infections (10%), ear infections, urinary tract infections, and gastroenteritis (5%) (Table 2). All isolates that were resistant to imipenem and amikacin were positive for both genes Metallo-β-lactamase gene Imipenemase (*bla-_IM_*_P_) and aminoglycoside 6′-N-acetyltransferase (*aac (6*′*)-Ib*) (Appendix A).

### 3.5. The Combined Effect of Amikacin and Imipenem against Resistant E. coli Using the Checkerboard Dilution Technique

High synergistic activity was shown by combinations with a high reduction in MICs, which ranged from two- to eight-fold in comparison to amikacin and imipenem alone, respectively (Table 3).

### 3.6. Time-Kill Studies

This test was conducted on three isolates: one isolate showing resistance to both imipenem and amikacin (isolate no. 3, wound source (W3)), an isolate resistant to imipenem only (W2), and an isolate resistant to amikacin only (W1). Regarding *E. coli* resistant to both amikacin and imipenem (W3), Figure 2 shows that at 0.5**×** MIC of the antibiotic combination count decreased the initial bacterial count (8.2 log_10_ CFU/mL) after 24 h to 7.46 log_10_ CFU/mL, and such a combination regimen was shown to be synergistic in comparison to each drug alone. At 1**×** MIC of the combination group, the colony count decreased to 4.7 log_10_ CFU/mL showing 3.5 log_10_ CFU/mL reductions which indicated a bactericidal and synergistic effect. At 2**×** MIC, there was a decrease in the bacterial count by 2.26 log_10_ CFU/mL reduction after 12 h, indicating a bacteriostatic effect and a decrease by 4.2 log_10_ CFU/mL reduction after 24 h, indicating bactericidal and synergistic activity in comparison to each drug alone.

At 4× MIC, the bacterial count decreased after 4 h to 4.6 log_10_ CFU/mL with 3.6 log_10_ CFU/mL reduction, indicating bactericidal activity. A significant difference between the four groups (*p* value ˂ 0.001) was shown at each concentration. Regarding *E. coli* resistant to imipenem (W2), the 0.5× MIC of combination group showed no significant effect (Figure 3). In the 1× MIC combination group, a reduction of 2.5 log_10_ CFU/mL was shown, indicating bacteriostatic activity after 8 h, and a reduction of 4.2 log_10_ CFU/mL was shown, indicating bactericidal and synergistic activity after 24 h. There was a statistically significant difference between the four groups (*p*-value ˂ 0.001) in each concentration. Regarding *E. coli* resistant to amikacin (W1), Figure 4 shows that at 0.5× MIC, the combination group showed no significant effect, while at 1× MIC combination, a decrease in count by 2.33 log_10_ CFU/mL reduction was observed, indicating a bacteriostatic effect after 8 h, and a reduction of 4.2 log_10_ CFU/mL was observed, representing a bactericidal and synergistic effect after 24 h. There was a significant difference between the four groups (*p* value ˂ 0.001) at each concentration (Appendix A).

### 3.7. In Vivo Studies

Figure 5 shows that the average blood bacterial counts were 9.41 log_10_ CFU/mL 3 h after infecting the mice (i.e., when the treatment started) with *E. coli* (W3*)*. The blood microbial count constantly increased in the control group (infected, untreated) to 10.36 log_10_ CFU/mL after 27 h using *E. coli*. Meanwhile, the blood microbial count reduced in all the infected treated groups using amikacin, imipenem, and a combination to 8.71 log_10_ CFU/mL, 8.25 log_10_ CFU/mL, and 6 log_10_ CFU/mL, respectively. After 27 h, the combination indicated bactericidal and synergistic activity. There was a significant difference among the tested groups (*p* < 0.001): control group and imipenem, control and amikacin, control and combination, amikacin and imipenem, amikacin and combination, and imipenem and combination (Appendix A). 

### 3.8. Gene Expression (Real-Time PCR) Results

Figure 6 shows the gene expression for selected *E. coli*-resistant isolates that were treated with amikacin and imipenem separately and in combination. The control samples showed no fold change for Metallo-β-lactamase gene Imipenemase (*bla-_IM_*_P_) or aminoglycoside 6′-N-acetyltransferase (*aac (6′)-Ib*). The isolate (W3) was selected to test the effect of the combination on the expression of genes using a real-time polymerase chain. The results show a decrease in the expression for the Metallo-β-lactamase gene Imipenemase (*bla-_IM_*_P_) gene equivalent to 9.1896 and the aminoglycoside 6′-N-acetyltransferase (*aac (6*′*)-Ib*) gene equivalent to 6.1475 after treatment with an amikacin/imipenem combination. The isolate (W1) and isolate (W2) which showed resistance to amikacin and imipenem, respectively, were selected to determine the expression of each gene using RT-PCR. Both isolates showed over-expression of Metallo-β-lactamase gene Imipenemase (*bla-_IM_*_P_) and aminoglycoside 6′-N-acetyltransferase (*aac (6*′*)-Ib*) in the untreated isolates rather than the treated isolates, confirming the importance of each gene on the resistance mechanism. The results also reveal that a combination 0.25× MIC of imipenem + 0.5× MIC of amikacin presented a higher decrease in expression than that showed by 0.5× MIC of imipenem +0.25× MIC of amikacin (Appendix A).

### 3.9. Scanning Electron Microscopy (SEM)

The effect of amikacin, imipenem, and their combination on the cell structure of *E. coli* (No.3) was confirmed by the SEM analysis in Figure 7A–D. In comparison to the control group, the cells treated with the tested antibiotics displayed a change in morphology in the form of cell elongation and swelling. The examined antibiotic-treated bacteria showed significant structural alterations on the outer membrane of *E. coli*, resulting in cell destruction, while untreated bacteria were intact (regular rod-shaped) and showed smooth surfaces. The outer membrane and cellular structure were altered by the combination of antibiotics.

## 4. Discussion

The emergence of bacterial resistance is considered a critical global issue. As infections by bacterial-resistant strains are associated with high morbidity and mortality, the World Health Organization (WHO) lists carbapenem-resistant strains, especially carbapenem-resistant Enterobacteriaceae (CRE), as one of the top tier of antibiotic-resistant “priority pathogens” that are considered as the greatest threat to human health [35,36]. Treatment with antibiotic combinations is recommended in severe infections caused by resistant strains [37,38,39,40]. In vitro and in vivo synergy tests can provide guidance for these combination therapies. This study was conducted to evaluate the effectiveness of the combination treatment of imipenem and amikacin as an example of beta-lactam and aminoglycoside antibiotics in the treatment of multidrug-resistant (MDR) *E. coli* using checkerboard, time-kill curve studies techniques, and in vivo studies. Our analysis showed that *E. coli* was the most common species (40%), which agreed with many studies [41,42]. A study by Giacobbe, et al. [43] showed that *Escherichia coli*, *Klebsiella pneumoniae*, *Pseudomonas aeruginosa*, and *Acinetobacter* spp. were the most common isolated pathogens that cause blood stream infections. On the other hand, Bhatt, et al. [44] reported a higher percentage of *E. coli* (50.4%). Our study indicated that multidrug-resistant (MDR) *E. coli* isolates were obtained from wounds, urinary tract infections, patients admitted to intensive care units, ear infections, burns, chest infections, and gastroenteritis. Antimicrobial susceptibility testing revealed that the most active antibiotics were piperacillin/tazobactam (33.3%) and gentamycin (34%). On the other hand, high resistance was observed against imipenem (61.6%), gentamycin (60%), and amikacin (58.3%). Similar results were shown by Dokla, et al. [45] and Vena, et al. [46]. Our results show that 80% and 66.7% of *E. coli* harbored *bla-_IMP_* and *aac(6′)-Ib*, respectively. Elbadawi, et al. [47] reported that 14.9% of *Escherichia coli* isolates harbored carbapenemase genes. Soliman, et al. [48] revealed that 33.8% Gram-negative bacteria were carbapenem-resistant GNB, including *E. coli* isolates (9.2%), *K. pneumoniae* isolates (3%), *P. stuartii* isolates (10.7%), and seven *P. aeruginosa* isolates (10.7%). Four *E. coli*, two *K. pneumoniae*, and seven *P. stuartii* were found to produce NDM-1, out of 1392.3% of strains; 12 were pan-aminoglycosides-resistant and had class 1 integron-carrying *aac(6′)-Ib*. 

Our study showed synergistic activity with the combination of imipenem and amikacin using a checkerboard assay where a combination of both drugs showed MIC values below the recognized breakpoints for resistance. In addition, the results obtained by time-kill experiments approved the synergistic activity of the tested combinations. The use of different antibiotic combinations was tested by Al-Tamimi, et al. [15] who reported positive synergistic activity by cephalosporins and amoxicillin/clavulanate with cefotaxime or cefixime compared to their combinations with cefpodoxime, cefdinir, and ceftazidime. Drago, et al. [17] reported that synergistic and additive effects were detected by combinations of levofloxacin and imipenem or amikacin, and between ciprofloxacin and amikacin against ESBL-producing *E. coli.* Loho, et al. [16] reported the effect of doripenem and amikacin on *Acinetobacter baumannii*, *Pseudomonas aeruginosa*, and *Klebsiella pneumoniae.* They reported that a synergistic effect only appeared in one isolate. Additive effects were found in 24 (35.3%) isolates, and indifferent interactions appeared in 43 (63.2%) isolates. They concluded that the combination of doripenem with amikacin significantly reduced MIC in all isolated strains when compared to the MIC of each antibiotic separately. Expression changes of Metallo-β-lactamase gene Imipenemase (*bla-_IM_*_P_) and aminoglycoside 6′-N-acetyltransferase (*aac (6*′*)-Ib*) genes are responsible for the production of amino-glycosidase and beta-lactamase enzymes. Our study observed a reduction in gene expression using qRT-PCR in the case of treatment with 0.25× MIC imipenem + 0.5× MIC amikacin compared to the effect of a single antibiotic. These changes in expression may be due to one of the mechanisms of synergy between aminoglycosides and β-lactams by the ability of β-lactams to increase the uptake of aminoglycosides [49]. The changes in the morphology of the tested isolates were confirmed by SEM in the presence and in the absence of the tested antibiotics. The SEM observations confirmed the morphological changes and disruption to the outer membrane of the tested isolates. The scanning electron microscopy showed a great disruption to the cells treated by aminoglycosides and β-lactams. Similar results were obtained by Yadav, et al. [50] and Hayami, et al. [51].

## 5. Conclusions

Amikacin/imipenem combinations were found to be a therapeutic option in controlling serious infections caused by multidrug-resistant (MDR) *E. coli* strains according to our results. This combination can also decrease the resistance risk of monotherapy while relieving the stress of clinical treatment.

## Figures and Tables

**Figure 1 tropicalmed-07-00281-f001:**
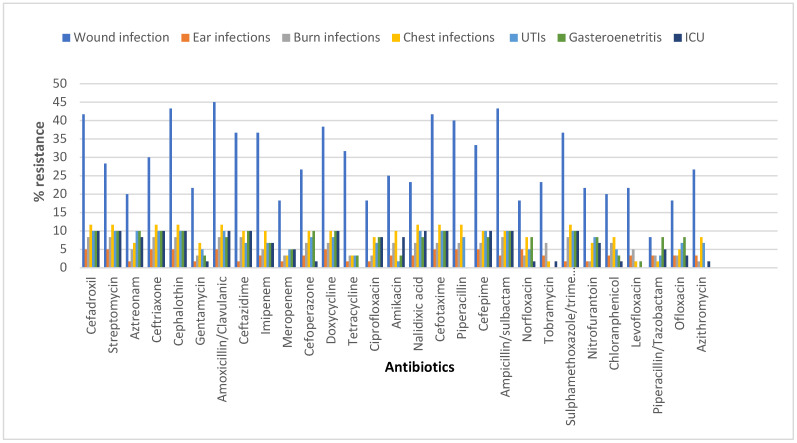
Antibiotic resistance profile of *E. coli* according to the infection type.

**Figure 2 tropicalmed-07-00281-f002:**
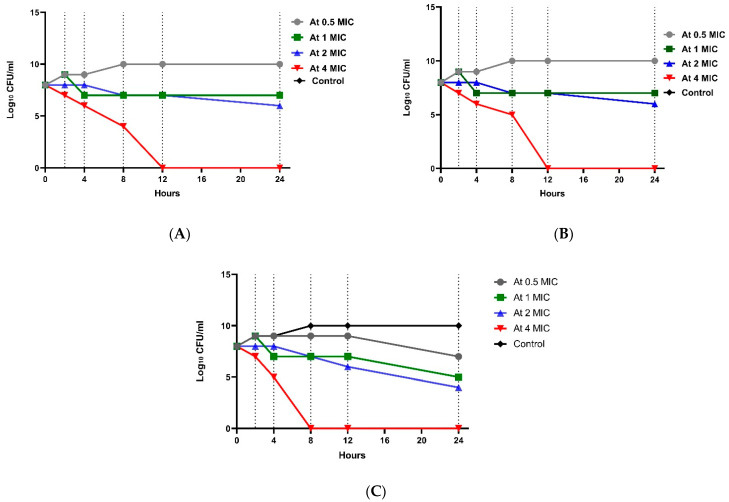
Time-kill assay for isolate no. 3 resistant to both imipenem and amikacin. (**A**): *E. coli* isolate treated with amikacin at different concentrations; (**B**): *E. coli* isolate treated with imipenem at different concentrations; and (**C**): *E. coli* isolate treated with a combination at different concentrations.

**Figure 3 tropicalmed-07-00281-f003:**
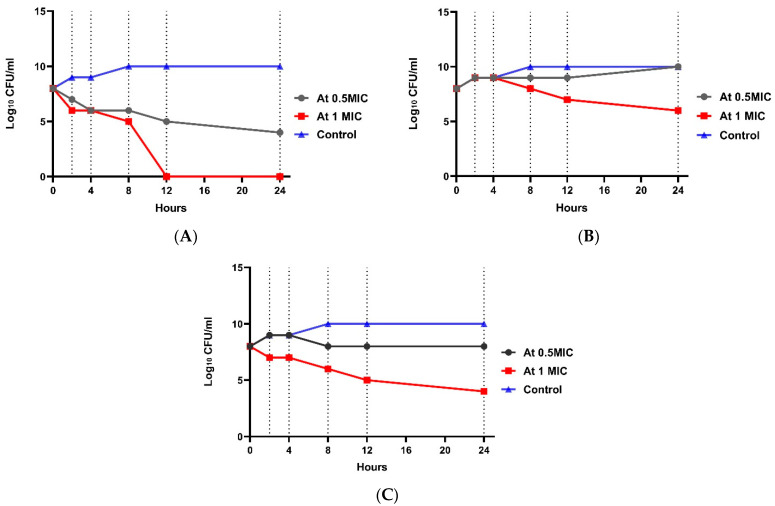
Time-kill assay for isolate no. 1 resistant to imipenem. (**A**): *E. coli* isolate treated with amikacin at different concentrations; (**B**): *E. coli* isolate treated with imipenem at different concentrations; and (**C**): *E. coli* isolate treated with a combination at different concentrations.

**Figure 4 tropicalmed-07-00281-f004:**
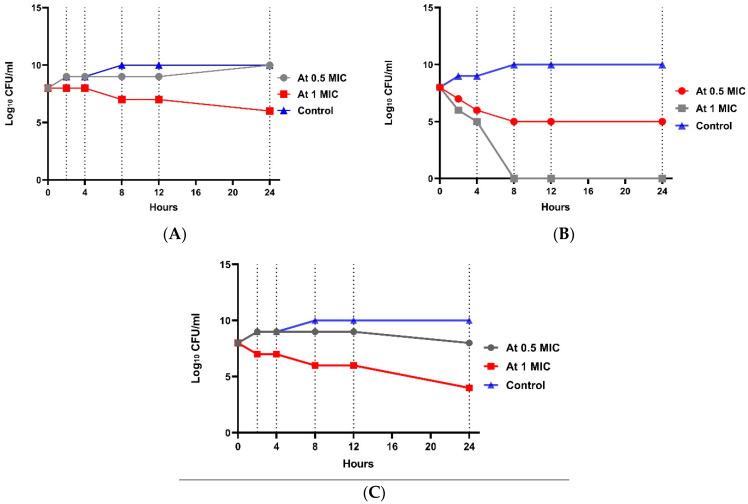
Time-kill assay for isolate no. 2 resistant to amikacin. (**A**): *E. coli* isolate treated with amikacin at different concentrations; (**B**): *E. coli* isolate treated with imipenem at different concentrations; and (**C**): *E. coli* isolate treated with a combination at different concentrations.

**Figure 5 tropicalmed-07-00281-f005:**
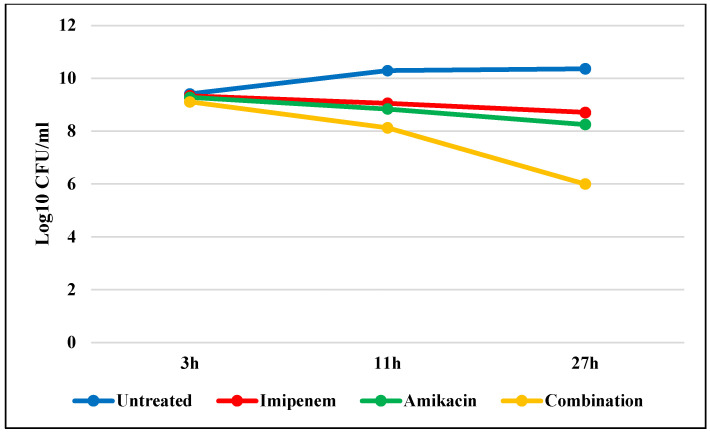
Average blood bacterial count of *E. coli* isolated from untreated animals and animals treated with imipenem, amikacin, and a combination at different times.

**Figure 6 tropicalmed-07-00281-f006:**
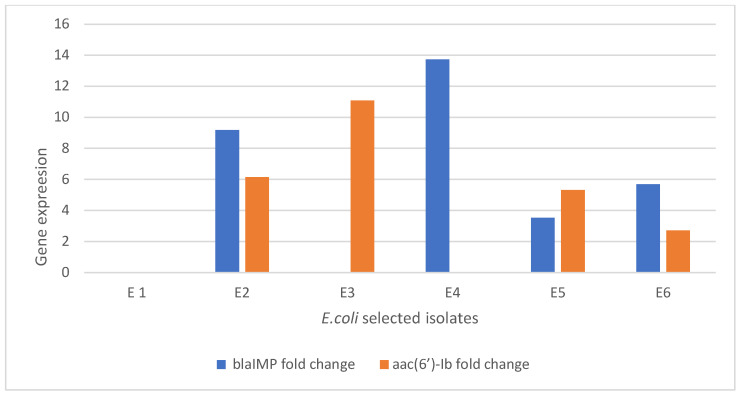
Gene expression of resistant *E. coli* after treatment: **E1**: control; **E2**: *E. coli* (w3) resistant to amikacin harboring aminoglycoside 6′-N-acetyltransferase (*aac (6*′*)-Ib*) and imipenem harboring Metallo-β-lactamase gene Imipenemase (*bla-_IM_*_P_); **E3**: *E. coli* (W1) resistant to amikacin harboring aminoglycoside 6′-N-acetyltransferase (*aac (6*′*)-Ib*) *only*; ***E4***: *E. coli* (W2) resistant to imipenem harboring Metallo-β-lactamase gene Imipenemase (*bla-_IM_*_P_) only; **E5**: *E. coli* isolate (w3) after treatment with 0.25 MIC of amikacin +0.5 MIC of imipenem; and **E6**: *E. coli* isolate (w3) after treatment with 0.25 MIC of imipenem +0.5 MIC of amikacin.

**Figure 7 tropicalmed-07-00281-f007:**
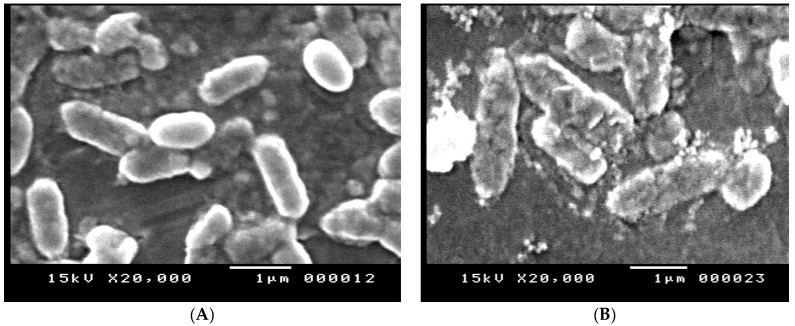
(**A**) SEM image of *E. coli* control, (**B**) SEM image of *E. coli* treated with imipenem, (**C**) SEM image of *E. coli* treated with amikacin, and (**D**) SEM image of *E. coli* treated with amikacin/imipenem- treated cells.

**Table 1 tropicalmed-07-00281-t001:** PCR primers.

Gene	Primer Sequence (5′-3′)	Annealing Temperature	Product Size (bp)	Reference
Metallo-β-lactamase gene Imipenemase (*bla-_IMP_*)	F:CATGGTTTGGTGGTTCTTGT	59	488	[27]
	R:ATAATTTGGCGGACTTTGGC			
Aminoglycoside 6′-N-acetyltransferase (*aac(6′)-Ib*)	F:AGTACTTGCCAAGCGTTTTAGCGC	51	365	[28]
	R:CATGTACACGGCTGGACCAT			
16S rRNA	F: GCTGACGAGTGGCGGACGGG	55	253	[29]
	R:TAGGAGTCTGGACCGTGTCT			

**Table 2 tropicalmed-07-00281-t002:** Distribution of Metallo-β-lactamase gene Imipenemase (*bla-_IM_*_P_) and aminoglycoside 6′-N-acetyltransferase (*aac (6*′*)-Ib*) genotype within 60 isolated *E. coli*.

Source of Infection	No. of Isolates	Metallo-β-lactamase Gene Imipenemase (*bla-_IMP_*) Positive Isolates No. (%) *^,^ **	Aminoglycoside 6′-N-acetyltransferase (*aac(6′)-Ib*) Positive Isolates No. (%) *^,^ **
Wound	27	23 (47.9%)	19 (47.5%)
Ear infections	3	2 (4.2%)	2 (5%)
Burn infections	5	3 (6.3%)	4 (10%)
Chest infections	7	6 (12.5%)	6 (15%)
Urinary tract infections	6	5 (10.4%)	2 (5%)
Gastroenteritis infections	6	4 (8.3%)	2 (5%)
ICU	6	5 (12.5%)	5 (12.5%)
Total	60	48 (80%)	40 isolates (66.7%)

* Percent correlated to total no. of positive isolates for each gene, ** percent correlated to total no. of *E. coli* of isolates.

**Table 3 tropicalmed-07-00281-t003:** Determination of the combined effect of amikacin and imipenem against resistant *E. coli* using checkerboard assay.

Name of Bacteria	MIC (µg/mL)				FIC_index_ 0.5× MIC Amikacin + 0.25× MIC Imipenem	FIC_index_ 0.25× MIC Amikacin + 0.5× MIC Imipenem	Outcome
	Amikacin Alone	Imipenem Alone	Combination of 0.5× MIC Amikacin + 0.25× MIC Imipenem	Combination of 0.25× MIC Amikacin + 0.5× MICImipenem			
*E. coli* (No.3)	1024	1024	32	32	0.06	0.06	Synergistic
*E. coli* (No.7)	1024	1024	32	32	0.06	0.06	Synergistic
*E. coli* (No.10)	1024	1024	32	32	0.06	0.06	Synergistic
*E. coli* (No.12)	1024	256	32	8	0.156	0.04	Synergistic
*E. coli* (No.13)	1024	256	32	8	0.156	0.04	Synergistic
*E. coli* (No.17)	1024	256	32	8	0.156	0.04	Synergistic
*E. coli* (No.19)	512	256	1	0.5	0.005	0.003	Synergistic
*E. coli* (No.21)	1024	256	32	8	0.156	0.26	Synergistic
*E. coli* (No.23)	1024	8	1	1	0.126	0.126	Synergistic
*E. coli* (No.25)	1024	8	1	1	0.126	0.126	Synergistic
*E. coli* (No.27)	512	256	1	0.5	0.0059	0.003	Synergistic
*E. coli* (No.29)	128	256	0.5	2	0.0059	0.023	Synergistic
*E. coli* (No.32)	128	256	0.5	2	0.0059	0.023	Synergistic
*E. coli* (No.35)	512	256	0.5	4	0.003	0.023	Synergistic
*E. coli* (No.36)	512	8	4	0.5	0.5	0.017	Synergistic
*E. coli* (No.37)	512	8	4	0.5	0.5	0.063	Synergistic
*E. coli* (No.39)	512	8	4	0.5	0.5	0.063	Synergistic
*E. coli* (No.41)	512	8	0.5	1	0.0634	0.13	Synergistic
*E. coli* (No.47)	1024	256	0.5	8	0.0024	0.04	Synergistic
*E. coli* (No.51)	512	256	8	4	0.047	0.023	Synergistic
*E. coli* (No.56)	1024	256	32	1	0.16	0.005	Synergistic
*E. coli* (No.57)	512	256	32	1	0.19	0.006	Synergistic
*E. coli* (No.59)	512	64	32	1	0.5	0.018	Synergistic
*E. coli* (No.60)	512	256	32	1	0.19	0.006	Synergistic

FIC index ≤ 0.5; antagonism showed as FIC index of ≥2, and additive showed as FIC index of >0.5 but ≤1.

## Data Availability

The data presented in this study are available in this article.

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
