# Peer review of "In Vitro and In Vivo Effect of Amikacin and Imipenem Combinations against Multidrug-Resistant E. coli"

_tropicalmed, 2022, doi:10.3390/tropicalmed7100281_

Round 1

Reviewer 1 Report

I appreciate the author’s effort in addressing the effect of combination against drug resistant E.coli. They used various in vitro and an in vivo model to investigate their hypothesis. However, I do not see any difference between this study and a recent study conducted by the same authors in P. aeruginosa except the bacteria which is another Gram negative bacteria i.e. E. coli. (Effect of Imipenem and Amikacin Combination against Multi-Drug Resistant Pseudomonas aeruginosa). Other researchers have also mad e the same study “Evaluation of Effectiveness of Antibiotic Combination Therapy in Multi Drug Resistant Escherichia Coli in Vitro and in Vivo (Nandita Banik*, 1 , Shamsuzzaman SM ; Fortune J Health Sci 2021; 4 (3): 468-476). It would have been better if the authors explained what is new and important for this study. In addition, the manuscript should be carefully revised for coherence, typo, and grammar. The results should also be properly discussed.

Abstract: consider reorganization of the abstract and define abbreviations when used for the first time.

Introduction: suggest dividing it into two or three paragraphs.

Methods: The authors did not mentioned anything about studies/isolation of other gram-negative bacteria in the methods section and it’s a little confusing to see some data on the results (Line 200).

It would have been better if the authors provide the resistance breakpoints for the tested antibiotics mentioned in the methods or the results.

Line 215-217 is already mentioned in the introduction and I do not feel it is necessary to repeat it here.

The results of kill study are a bit confusing. It would have been betters if the authors avoid unnecessary details that can easily be found on the plots. It is difficult to understand the difference between Figure 2, 3 and 4 as it stands.

The resolution of SEM figures are all different and it’s misleading. As it stands, it seems that Amikacin (panel C) has significant effect on bacterial morphology than others, in contrary to the author’s statement on line 313.

Line 328. “Our analysis 328 showed that E. coli was the most common species (40%).” Common in terms of what?

Line 338-341: It would have been better if the authors explain the findings of previous studies in percent (to be consistent)

Line 351-353…in which type of bacteria?

Please make a sound discussion and appropriate conclusion.

Good luck!

Author Response

I appreciate the author’s effort in addressing the effect of combination against drug resistant E.coli. They used various in vitro and an in vivo model to investigate their hypothesis. However, I do not see any difference between this study and a recent study conducted by the same authors in P. aeruginosa except for the bacteria which is another Gram-negative bacteria i.e. E. coli. (Effect of Imipenem and Amikacin Combination against Multi-Drug Resistant Pseudomonas aeruginosa).

Answer:

In our area, P. aeruginosa and E. coli are the most common microorganisms in our hospitals and are the main cause of life-threatening infections. Also, their un a controllable prescription for carbapenem antibiotics and some antibiotics like amikacin and vancomycin. So, we decide to study the effect of imipenem and amikacin against Gram-negative bacteria

It would have been better if the authors explained what is new and important for this study. In addition, the manuscript should be carefully revised for coherence, typo, and grammar. The results should also be properly discussed.

Answer:

It was revised and corrected

Other comments

  • Abstract: consider reorganization of the abstract and define abbreviations when used for the first time.

Answer

Done in the manuscript

  • Introduction: suggest dividing it into two or three paragraphs.

       Answer

       Done in the manuscript

  • Methods: The authors did not mentioned anything about studies/isolation of other gram-negative bacteria in the methods section and it’s a little confusing to see some data on the results (Line 200).

Answer

Added in the manuscript

  • It would have been better if the authors provide the resistance breakpoints for the tested antibiotics mentioned in the methods or the results.

Answer

Resistance break point as reported according to CLSI 2018 as mentioned in the manuscript.

  • Line 215-217 is already mentioned in the introduction and I do not feel it is necessary to repeat it here.

Answer

              Removed from the manuscript

  • The results of kill study are a bit confusing. It would have been betters if the authors avoid unnecessary details that can easily be found on the plots. It is difficult to understand the difference between Figure 2, 3 and 4 as it stands.

Answer

In time kill study the study selected three different isolates their sources were wound infection as an example

  1. 1st isolate was E.coli resistant for both drugs imipenem and amikacin (figure 2) , this figure showed the effect of imipenem at different concentrations (i.e 0.5xMIC, 1xMIC ,2XMIC and4XMIC) and at different times (0hr, 2hr,4hr,8hr,12hr and 24hr),also showed the effect of amikacin  at different concentrations (i.e 0.5xMIC, 1xMIC ,2XMIC and4XMIC) and at different times (0hr, 2hr,4hr,8hr,12hr and 24hr),and the effect of combination which showed better effect on bacterial count at the same concentrations and same times.
  2. 2nd isolate E.coli resistant to imipenem only(figure 3) this figure showed the effect of imipenem at different concentrations (i.e 0.5xMIC, and 1xMIC) and at different times (0hr, 2hr,4hr,8hr,12hr and 24hr),also showed the effect of amikacin at different concentrations (i.e 0.5xMIC, and 1xMIC) and at different times (0hr, 2hr,4hr,8hr,12hr and 24hr),and the effect of combination which showed better effect on bacterial count at the same concentrations and same times.

  1. 3rd isolate E.coli resistant to amikacin only (figure 4) this figure showed the effect of imipenem at different concentrations (i.e 0.5xMIC, and 1xMIC) and at different times (0hr, 2hr,4hr,8hr,12hr and 24hr),also showed the effect of amikacin at different concentrations (i.e 0.5xMIC, and 1xMIC) and at different times (0hr, 2hr,4hr,8hr,12hr and 24hr),and the effect of combination which showed better effect on bacterial count at the same concentrations and same times.

 N.B Figure (3&4) the study used two concentrations only(i.e 0.5xMIC, and 1xMIC) as the selected isolates one was resistant to imipenem only and the other isolate was resistant to amikacin only and the second reason that there wasn’t any bacterial count at 2XMIC and 4XMIC

  • The resolution of SEM figures are all different and it’s misleading. As it stands, it seems that Amikacin (panel C) has significant effect on bacterial morphology than others, in contrary to the author’s statement on line 313.

Answer

Changed in the manuscript

  • Line 328. “Our analysis 328 showed that E. coli was the most common species (40%).” Common in terms of what?

Answer

  1. coli was the most common species (40%) of total Gram-negative bacteria (150 isolate)

i.e (60/150)

  • Line 338-341: It would have been better if the authors explain the findings of previous studies in percent (to be consistent)

Answer

Done in the manuscript

  • Line 351-353…in which type of bacteria?

Answer

Type of bacteria was E.coli the reference (Al-Tamimi, M.; Abu-Raideh, J.; Albalawi, H.; Shalabi, M.; Saleh, S. Effective Oral Combination Treatment for Extended-Spectrum Beta-Lactamase-Producing Escherichia coli. Microb Drug Resist 2019, 25, 1132-1141, doi:10.1089/mdr.2019.0065)

Reviewer 2 Report

General comments

The manuscript requires language editing.

Please remove all the subheadings from the Abstract

Specific comments

Line 16-17:...E. coli has spread worldwide...

Line 18: ... infections. The current study aimed to evaluate the in-vitro...

 Line 22: Scanning electron microscope was used to.....

Line 26:...E. coli isolates harboring...

Line 27: ...was 48 and 40, respectively, most of which were isolated from wound

Line 33: Write MDR and XDR in full.

Line 38: Specify that "E. coli pathotypes" because not all E. coli cause infection

Line 42: Please rephrase. E. coli is already a species. Maybe you want to talk about strains

Line 47: Be consistent with the spelling of multidrug

Line 58: carbapenem-resistant

Line 59-60: "As, these resistant strains can be treated by polymyxins, tigecycline, fosfomycin, and aminoglycosides."  This sentence is hanging and not clear

Linee 60-62: Please cite at least four of those "many studies"

Line 73: be consistent with "in-vitro" and "in-vivo" writing stye.

Line 79: hospital

Line 79-81: suffering from chest, wound, burn, urinary tract, and ear infections and gastroenteritis.

Line 81-82: This sentence is hanging

Line 83-84: Please describe E. coli's colony morphology on EMB properly. The colonies do not appear as green metallic sheen.....the colony appears with a green metallic sheen

Line 92: Delete "Bauer et al."

Line 219: harboring the 

Line 227:  were positive for...why in bold?

Author Response

General comments

The manuscript requires language editing.

  • Please remove all the subheadings from the Abstract

Answer

Done in the manuscript

Specific comments

  • Line 16-17:...E. coli has spread worldwide...

Answer

Done in the manuscript

  • Line 18: ... infections. The current study aimed to evaluate the in-vitro...

Answer

Done in the manuscript

  • Line 22: Scanning electron microscope was used to.....

Answer

Done in the manuscript

  • Line 26:... coli isolates harboring...

Answer

Done in the manuscript

  • Line 27: ...was 48 and 40, respectively, most of which were isolated from wound

Answer

Done in the manuscript

  • Line 33: Write MDR and XDR in full.

Answer

Done in the manuscript

  • Line 38: Specify that "E. coli pathotypes" because not all E. coli cause infection

Answer

Done in the manuscript

  • Line 42: Please rephrase. coli is already a species. Maybe you want to talk about strains

Answer

Done in the manuscript

  • Line 47: Be consistent with the spelling of multidrug

Answer

Done in the manuscript

  • Line 58: carbapenem-resistant

Answer

Done in the manuscript

  • Line 59-60: "As, these resistant strains can be treated by polymyxins, tigecycline, fosfomycin, and aminoglycosides." This sentence is hanging and not clear

Answer

Removed from manuscript

  • Linee 60-62: Please cite at least four of those "many studies"

Answer

Done in the manuscript

  • Line 73: be consistent with "in-vitro" and "in-vivo" writing stye.

Answer

Done in the manuscript

  • Line 79: hospital

Answer

Done in the manuscript

  • Line 79-81: suffering from chest, wound, burn, urinary tract, and ear infections and gastroenteritis.

Answer

Done in the manuscript

  • Line 81-82: This sentence is hanging

Answer

Removed from manuscript

  • Line 83-84: Please describe E. coli's colony morphology on EMB properly. The colonies do not appear as green metallic sheen.....the colony appears with a green metallic sheen
  • Line 92: Delete "Bauer et al."

Answer

Removed from manuscript

  • Line 219: harboring the

Answer

Done in the manuscript

  • Line 227: were positive for...why in bold?

Answer

Done in the manuscript

Reviewer 3 Report

In this manuscript, the use of the combination of amikacin and imipenen against strains of Multi-Drug Resistant E. coli is studied.

The following comments are made.

1. Exhaustive review of English.

2. Line 22. “used to detect bla-IMP and aac (6')-Ib”. Indicate what they are. Review throughout the text

3. Lines 26-27. 48 and 40 of how many? Better put a percentage.

4. Line 33. What do MDR and XDR mean? Put the meaning of abbreviations first. Review throughout the text.

5. Line 37: Gram is capitalized. Review throughout the text.

6. Line 38. The term “flora” is no longer used, it is microbiota. Correct throughout the text.

7. Line 44. What does MDR mean?

8. Line 50. “arethe”. Correct

9. Line 53: “Ambler Class A”. What do you mean?

10. Line 66. What does MIC mean?

11. Line 78. Describe the group of patients: ages, gender?

12. Line 81. How did you take the samples?

13. Line 82. “Samples were processed and grown”. How were the samples processed?

14. Lines 79, 88-89. Capitalized proper nouns.

15. Line 98. Put the same font.

16. Line 103. Reference 22 is not from CLSI, put correct reference.

17. Line 104. “molecular detection of bla-IMP and aac (6’)-Ib”. Indicate what they are? Correct throughout the text.

18. Table 1. “Aac(6’)-Ib” and “16s”. Correct is "aac" and "16S".

19. Line 112. Remove bold.

20. Line 116. How did you calculate the FIC index? explain it.

21. Line 125. How did you do the Bacterial counts?

22. Line 135. What does ICR mean?

23. Did you request authorization to handle animals?

24. Line 155: “200ul”. Why in red? Put mL. Correct throughout the text.

25. Line 173. CT is capitalized

26. Figure 1 Cannot see the result bars. Correct.

27. To consider a resistant strain, what are the established MIC values. Put it in Material and methods.

28. Line 223-227. remove bold

29. Table 3. Why do you have two values ​​when combining antibiotics? Also, explain at the foot of the table what the different FICs mean.

30. Figure 2. MIC is capitalized. Correct in all text including supplementary files. Correct mL

31. Line 258. What does AK mean? Put the meaning of the abbreviations first. Review throughout the text.

32. Lines 238-260. Homogenize, put mL not ml. Review throughout the text and figures.

33. What is the real difference between Figures 2, 3 and 4? If there is no difference, leave only one Figure.

34. Line 262. What does S4, S5, S6 mean?

35. Figure 5. Correct gene nomenclature is italicized and lowercase.

36. Figure 7. Use the same magnification for all cases.

37. Line 320. Proper nouns first letter capitalized. Review throughout the text.

38. Lines 323-325. If there are already combination studies of the antibiotics used. What is the novelty of your study. Explain it.

39.Line 339-340. Remove italics from words.

40. Discuss the combination of antibiotics used and antibiotic resistance, what happens?

41. What dose do you recommend to treat infections?

42. References: Homogenize the name of the Journal. Is it abbreviated or complete? In some cases you put it complete and in others abbreviated. Consult the authors guide.

Round 2

Reviewer 1 Report

Thank you for responding to my review. However, some of my concerns are not properly addressed. Ex: Grammar, typo and coherence. Gene names should be following the standard nomenclature and should also be italicized

1.       Abstract: Lines 16-18, 20-23, 29-30 should be revised!

2.       Line 57..Suggest replace “cause” with appropriate word such as  “mechanism” or any other. “Cause” is most of the time related with irrational antibiotic use or others that leads to resistance in the host.

3.       Line 109. Suggest revising the subheading as  “PCR detection of of bla-IMP and aac (6’)-Ib genes”

4.       Line 144..What do you mean  by “Old specific pathogen free”?

5.       Line 147.Please include injection volume

6.       Line 166 “tested group received intraperitoneal combination of amikacin and imipenem the same as a single dose” would be better if the authors clarify this. Does this mean that the mice were injected with 400 ul (200 for each)? 

7.       Please provide appropriate legend for both  figures on page 6

8.       I believe the time kill result (text and figures) could be explained in a simplest way. Suggest removing the vertical (dotted lines) on the graph. No need to mention E. coli everywhere on the graph because your study is only in E. coli

9.       The SEM figure is still misleading because, I stated out in the first revision, the figures do not have equal resolution. For examples, the bacteria in Figure A are a bit elongated than those in C. Please provide a figure with better quality or remove this section form the manuscript.

1.   Line 351…”…. which was in agreement with many studies”.. at least 2 citations are needed.

I It would also be better if the authors discuss their findings systematically.

Good luck!

Author Response

Reviewer#1

Thank you for responding to my review. However, some of my concerns are not properly addressed. Ex: Grammar, typo, and coherence.

 Manuscript was revised

Gene names should be following the standard nomenclature and should also be italicized

Answer

Done in the manuscript

  1. Abstract: Lines 16-18, 20-23, 29-30 should be revised

It was revised and appeared with blue color of track changes.

  1. Line 57. Suggest replacing “cause” with an appropriate word such as “mechanism” or any other. “Cause” is most of the time related with irrational antibiotic use or others that leads to resistance in the host.

Answer

Corrected in the manuscript in line 63

  1. Line 109. Suggest revising the subheading as “PCR detection of of bla-IMP and aac (6’)-Ib genes”

Answer

Corrected in the manuscript

  1. Line 144..What do you mean by “Old specific pathogen free”?

Answer

Specific pathogen-free female ICR mice meant that they weren’t infected with any pathogen or bacteria. They were 6 weeks old. (The sentence was rewrite in the manuscript)

  1. 5. Line 147.Please include injection volume

Bacteria Mice were administered (0.5MacCfarland=1.5x108CFU/mL);0.2mL volume line 163

Drugs Antibiotic treatment was administered 0.2 mL volumes (based on the body weight) we use 200µl (100 µl each) (Mathe, Szabo et al. 2007).

  1. Line 166 “tested group received intraperitoneal combination of amikacin and imipenem the same as a single dose” would be better if the authors clarify this. Does this mean that the mice were injected with 400 ul (200 for each)?

Answer

Antibiotic treatment was administered 0.2 mL volumes (based on the body weight) we use 200µl (100 µl each) (Mathe, Szabo et al. 2007).

  1. Please provide appropriate legend for both figures on page 6

Answer

Changed in the manuscript

  1. I believe the time kill result (text and figures) could be explained in a simplest way. Suggest removing the vertical (dotted lines) on the graph. No need to mention E. coli everywhere on the graph because your study is only in E. coli

Answer

vertical (dotted lines) on the graph indicated change in number of CFU/mL at different times. Also, according to first round revision of one reviewer, doted lines added to show the time clearly on the curve.

  1. The SEM figure is still misleading because, I stated out in the first revision, the figures do not have equal resolution. For examples, the bacteria in Figure A are a bit elongated than those in C. Please provide a figure with better quality or remove this section form the manuscript.

Answer

All the figures had resolution 15kUx20.000, Figure A changed in the manuscript

  1. Line 351…”…. which was in agreement with many studies”.. at least 2 citations are needed.

Answer

Done in the manuscript

Reviewer 3 Report

Several of the comments were only made in the letter to the reviewer but not included in the manuscript. The comments are to be included in the manuscript and the work is better understood.

Author Response

Reviewer#3

Several of the comments were only made in the letter to the reviewer but not included in the manuscript. The comments are to be included in the manuscript and the work is better understood.

All comments were cited in the manuscript with track changes and highlights.

  1. Exhaustive review of English.

  1. 2. Line 22. “used to detect bla-IMP and aac (6')-Ib”. Indicate what they are. Review throughout the text

Answer

Done in the manuscript highlighted in line 23

  1. Lines 26-27. 48 and 40 of how many? Better put a percentage.

Answer

it was added at line 52 and a new reference was added (Done in the manuscript)

  1. Line 33. What do MDR and XDR mean? Put the meaning of abbreviations first. Review throughout the text.

Answer

 MDR was put as multi-drug resistant. But XDR is not present in the manuscript, and it means extensive drug resistance. Done in the manuscript

  1. 5. Line 37: Gram is capitalized. Review throughout the text.

Answer

Done in the manuscript

  1. 6. Line 38. The term “flora” is no longer used, it is microbiota. Correct throughout the text.

Answer

Done in the manuscript line 46 and highlighted

  1. Line 44. What does MDR mean?

It means multi-drug resistant, and it was added in manuscript

  1. Line 50. “arethe”. Correct

Answer

It was corrected in the manuscript

  1. line 53. “Amblar class A” what do you mean?

According to Ambler classification, class A is clavulanic acid-inhibitory extended-spectrum β-lactamases it was added in the manuscript with its reference.

  1. line 66. What does MIC mean?

Minimum inhibitory concentration was added in line 81

  1. Line 78. Describe the group of patients: gender and age?

Age range and gender were added at line 96

  1. Line 81. How did you take samples?

By sterile swabs for wound, burn and ear discharge and place it in the sterile cover containing sterile saline to avoid dryness till culture.

Sputum and urine were taken in sterile cups

Stool samples were collected in cups that were not necessary to be sterile.

  1. Line 82. “Samples were processed and grown” How were the samples processed?

Each type of samples were processed according to Collee, et al. [1]

  Urine samples

Urine samples patients were collected in a sterile, dry and wide necked container to collect midstream urine (MSU).

        Urine specimens were centrifuged at 3,000 r.p.m. for 20 minutes and supernatants were decanted. The pellets were streaked onto plates of Nutrient agar, MacConkey agar, Blood agar and Cetrimide agar. All inoculated plates were incubated aerobically at 37°C for 24 hours. Growth was examined both microscopically and biochemically.

wound swabs

For collecting wound exudates, abscess exudates and burn exudates special care was taken to avoid contamination of the specimens with commensal organisms by cleaning the skin around the wound with povidone-iodine for one minute. A sterile cotton swab was used to collect pus from the infected site was streaked on Nutrient agar, MacConkey agar, Blood agar and Cetrimide agar. All inoculated plates were incubated aerobically at 37°C for 24 hours. Growth was examined both microscopically and biochemically.

Respiratory tract samples

Concerning sputum specimens’ early morning sputum samples were collected. The patient was given a clean, dry, and wide necked and leak proof container and asked to cough deeply to produce a sputum specimen. All specimens were cultured within 4 hours of collection. They were streaked on Nutrient agar, MacConkey agar, Blood agar and Cetrimide agar. All inoculated plates were incubated aerobically at 37°C for 24 hours. Growth was examined both microscopically and biochemically.

Ear discharge samples

Ear swabs were collected from different patients by using sterile cotton swabs to collect pus from infected site. They were streaked on Nutrient agar, MacConkey agar, Blood agar and Cetrimide agar. All inoculated plates were incubated aerobically at 37°C for 24 hours. Growth was examined both microscopically and biochemically.

Stool samples

Stool samples were collected in clean card cups. About 5 g of each specimen was transferred into final container containing glycerol phosphate buffer (0.33M). The buffer included in the container to protect delicate pathogen against pH variation due to the temperature drop after stool being passed.

The entire specimen was transferred to the microbiological laboratory within 2 hours and was processed as soon as possible on the same day. Stool samples were firstly cultivated on blood agar, then streaked on MacConkey agar that support the growth of Enterobacteriaceae, on Cetrimide agar for the selective growth of Pseudomonas aeruginosa. After 24 hrs. incubation at 37°C, colonies obtained were examined microscopically and biochemically.

  1. Lines 79, 88-89. Capitalized proper nouns.

Answer

Done in the manuscript

  1. Line 98. Put the same font.

Answer

Done in the manuscript

  1. Line 103. Reference 22 is not from CLSI, put correct reference.

Answer

Done in the manuscript the reference was corrected to (reference26)

 Wayne, PA. "Clinical and Laboratory Standards Institute. Performance Standards for Antimicrobial Susceptibility Testing."  (2018).

  1. Line 104. “molecular detection of bla-IMP and aac (6’)-Ib”. Indicate what they are? Correct throughout the text.

Answer

Full name names were written in line 122.

  1. Table 1. “Aac(6’)-Ib” and “16s”. Correct is "aac" and "16S".

Answer

It was corrected

  1. Line 112. Remove bold.

Answer

It was removed

  1. Line 116. How did you calculate the FIC index? explain it.

Done in the manuscript from line 139-142

  1. Line 125. How did you do the Bacterial counts?

https://mospace.umsystem.edu/xmlui/bitstream/handle/10355/69341/5- VPC%20Viable%20Plate%20Count.pdf?sequence=22&isAllowed=y

  1. Line 135. What does ICR mean?

The ICR mouse is a strain of albino mice originating in SWISS and selected by Dr. Hauschka to create a fertile mouse line. Because mice of this strain have been sent to various places from the Institute of Cancer Research in the USA, the strain was named ICR after the initial letters of the institute

  1. Did you request authorization to handle animals?

The study protocol conformed to the ethical guidelines of the 1975, Declaration of Helsinki, as

revealed in a priori approval (8/2021) by Ethical review board of faculty of pharmacy, Deraya

university, Egypt as mentioned in the manuscript line 187-189

  1. Line 155: “200ul”. Why in red? Put mL. Correct throughout the text.

It was corrected throughout the text

  1. Line 173. CT is capitalized

Answer

It was corrected

  1. Figure 1 Cannot see the result bars. Correct.

Answer

Due to large number of tested antibiotics. It was enlarged

  1. To consider a resistant strain, what are the established MIC values. Put it in Material and methods.

Answer

We take tables containing Antibiotics included in CLSI as a guide in the interpretation. MICs for imipenem and amikacin were added in line 119

  1. Line 223-227. remove bold

Answer

It was removed

  1. Table 3. Why do you have two values ​​when combining antibiotics? Also, explain at the foot of the table what the different FICs mean.

Answer

2 values were used as we use 2 antibiotics which differ in their MICs. FICs values were added to the foot off the table

  1. Figure 2. MIC is capitalized. Correct in all text including supplementary files. Correct mL

Answer

All time kill assay curves replaced by more clear figures. (Figure2-Figure 4)

Correction was done

  1. Line 258. What does AK mean? Put the meaning of the abbreviations first. Review throughout the text.

Answer

It means amikacin. corrected throughout the text.

  1. Lines 238-260. Homogenize, put mL not ml. Review throughout the text and figures.

Answer

It was corrected

  1. What is the real difference between Figures 2, 3 and 4? If there is no difference, leave only one Figure.

In time kill study the study selected three different isolates their sources were

wound infection as an example

  1. 1st isolate was E.coli resistant for both drugs imipenem and amikacin (figure 2) , this

figure showed the effect of imipenem at different concentrations (i.e 0.5xMIC, 1xMIC

,2XMIC and4XMIC) and at different times (0hr, 2hr,4hr,8hr,12hr and 24hr), also showed

the effect of amikacin at different concentrations (i.e 0.5xMIC, 1xMIC ,2XMIC

and4XMIC) and at different times (0hr, 2hr,4hr,8hr,12hr and 24hr), and the effect of

the combination which showed a better effect on the bacterial count at the same concentrations and same times.

  1. 2nd isolate E.coli resistant to imipenem only(figure 3) this figure showed the effect of

imipenem at different concentrations (i.e 0.5xMIC, and 1xMIC) and at different times

(0hr, 2hr,4hr,8hr,12hr and 24hr), also showed the effect of amikacin at different

concentrations (i.e 0.5xMIC, and 1xMIC) and at different times (0hr, 2hr,4hr,8hr,12hr

and 24hr), and the effect of combination which showed a better effect on the bacterial count at

the same concentrations and same times.

  1. 3rd isolate E.coli resistant to amikacin only (figure 4) this figure showed the effect of

imipenem at different concentrations (i.e 0.5xMIC, and 1xMIC) and at different times

(0hr, 2hr,4hr,8hr,12hr and 24hr), also showed the effect of amikacin at different

concentrations (i.e 0.5xMIC, and 1xMIC) and at different times (0hr, 2hr,4hr,8hr,12hr

and 24hr), and the effect of combination which showed a better effect on the bacterial count at

the same concentrations and same times.

N.B Figure (3&4) the study used two concentrations only (i.e 0.5xMIC, and 1xMIC) as

the selected isolates one was resistant to imipenem only and the other isolate was

resistant to amikacin only and the second reason that there wasn’t any bacterial count at

2XMIC and 4XMIC

  1. Line 262. What does S4, S5, S6 mean?

S4: Supplement data table 4

S5: Supplement data table 5

S6: Supplement data table 6

  1. Figure 5. Correct gene nomenclature is italicized and lowercase.

Answer

It was corrected

  1. Figure 7. Use the same magnification for all cases.

Answer

Done in the manuscript

  1. Line 320. Proper nouns first letter capitalized. Review throughout the text.

Answer

Done in the manuscript

  1. Lines 323-325. If there are already combination studies of the antibiotics used. What is the novelty of your study. Explain it.

1- Imipenem and amikacin were commonly used antibiotics in hospitals but we noticed resistance against these drugs , and one of their mechanism of resistant was the presence of blaIMP and aac(6’)-Ib, combination of these drugs in our study revealed that they had synergistic effect by both in-vitro and in-vivo and both drugs in combination showed significant decrease of bacterial count.

2- There is a lack of clinical studies evaluating the efficacy of imipenem and amikacin combination, and there are only very limited cases published about the effect of carbapenem plus aminoglycoside combination therapy (Mathe, Szabo et al. 2007).

3- (Mathe, Szabo et al. 2007) reported that imipenem and amikacin are effective as a single

treatment in infections with ESBL-producing bacteria, which was opposite to our finding that revealed combination of these drugs was the best in decreasing bacterial count, so we believe in the future they will be the best choice to overcome this problem.

39.Line 339-340. Remove italics from words.

Answer

Done in the manuscript

  1. Discuss the combination of antibiotics used and antibiotic resistance, what happens?

Combination was between amikacin as example for aminoglycosides and between imipenem as an example of beta-lactams antibiotics (these drugs are commonly used in our hospitals and there was a significant resistance observed against these drugs, where the study detected the reason for this resistance was the presence of resistance genes blaIMP and aac(6’)-Ib which detected by conventional PCR ) , by applying this combination invitro by time killing curve and by checkerboard assay we found that bacterial count decreased significantly (i.e there was a synergistic effect), then the study observed that the gene expression of both resistant genes decreased by combination , after that the study confirmed this synergistic effect between both drugs by in vivo study (Bacterial count decreased significantly by combining imipenem and amikacin by invitro and in vivo studies).

  1. What dose do you recommend to treat infections?

Mice were treated by intraperitoneal injections with amikacin 15 mg/kg in every 8 h or imipenem 40 mg/kg in every 4 h or amikacin and imipenem in combination (doses and intervals were the same as in monotherapy) (Mathe, Szabo et al. 2007).

The study protocol conformed to the ethical guidelines of the 1975, Declaration of Helsinki, as revealed in a priori approval (8/2021) by Ethical review board of faculty of pharmacy, Deraya university, Egypt.

  1. References: Homogenize the name of the Journal. Is it abbreviated or complete? In some cases you put it complete and in others abbreviated. Consult the authors guide.

References were according to endnote (MDPI )

References:

Collee, John Gerald, Thomas Jones Mackie, and James Elvins McCartney. Mackie & Mccartney Practical Medical Microbiology: Harcourt Health Sciences, 1996.

Benson, H. J. (1967). "Microbiological applications; a laboratory manual in  eneral icrobiology."Cruickshank, R., J. Duguid, B. Marmion and R. Swain (1973). "Medical microbiology; a guide to the laboratory diagnosis and control of infection-v. 1: Microbial infections.-v. 2: The practice of medical microbiology-12

"Hedberg, C. W. and M. T. Osterholm (1993). "Outbreaks of food-borne and waterborne viral gastroenteritis." Clinical microbiology reviews 6(3): 199-210.

Mathe, A., D. Szabo, P. Anderlik, F. Rozgonyi and K. Nagy (2007). "The effect of amikacin and imipenem alone and in combination against an extended-spectrum beta-lactamase-producing Klebsiella pneumoniae strain." Diagn Microbiol Infect Dis 58(1): 105-110.

Round 3

Reviewer 1 Report

Dear authors,

I hope you will go through the manuscript carefully during the proof read stage (for typo, coherence and gene nomenclature)

Good luck!